# Mixed Infections Unravel Novel HCV Inter-Genotypic Recombinant Forms within the Conserved IRES Region

**DOI:** 10.3390/v16040560

**Published:** 2024-04-03

**Authors:** Natalia Echeverría, Fabiana Gámbaro, Stéphanie Beaucourt, Martín Soñora, Nelia Hernández, Juan Cristina, Gonzalo Moratorio, Pilar Moreno

**Affiliations:** 1Laboratorio de Virología Molecular, Centro de Investigaciones Nucleares, Facultad de Ciencias, Universidad de la República, Montevideo 11400, Uruguay; necheverria@fcien.edu.uy (N.E.); fagambaro3@gmail.com (F.G.); msonora@pasteur.edu.uy (M.S.); juan.cristinag@gmail.com (J.C.); moratorio@pasteur.edu.uy (G.M.); 2Laboratorio de Evolución Experimental de Virus, Institut Pasteur de Montevideo, Montevideo 11400, Uruguay; 3Viral Populations and Pathogenesis Laboratory, Institut Pasteur, 75015 Paris, France; stephanie.beaucourt@pasteur.fr; 4Laboratorio de Simulaciones Biomoleculares, Institut Pasteur de Montevideo, Montevideo 11400, Uruguay; 5Clínica de Gastroenterología, Hospital de Clínicas, Facultad de Medicina, Universidad de la República, Montevideo 11600, Uruguay; hernandez.nelia@gmail.com

**Keywords:** HCV, inter-genotypic recombination, IRES, NGS, mixed infections

## Abstract

Hepatitis C virus (HCV) remains a significant global health challenge, affecting millions of people worldwide, with chronic infection a persistent threat. Despite the advent of direct-acting antivirals (DAAs), challenges in diagnosis and treatment remain, compounded by the lack of an effective vaccine. The HCV genome, characterized by high genetic variability, consists of eight distinct genotypes and over ninety subtypes, underscoring the complex dynamics of the virus within infected individuals. This study delves into the intriguing realm of HCV genetic diversity, specifically exploring the phenomenon of mixed infections and the subsequent detection of recombinant forms within the conserved internal ribosome entry site (IRES) region. Previous studies have identified recombination as a rare event in HCV. However, our findings challenge this notion by providing the first evidence of 1a/3a (and vice versa) inter-genotypic recombination within the conserved IRES region. Utilizing advanced sequencing methods, such as deep sequencing and molecular cloning, our study reveals mixed infections involving genotypes 1a and 3a. This comprehensive approach not only confirmed the presence of mixed infections, but also identified the existence of recombinant forms not previously seen in the IRES region. The recombinant sequences, although present as low-frequency variants, open new avenues for understanding HCV evolution and adaptation.

## 1. Introduction

Hepatitis C virus (HCV) is a major global health challenge, affecting an estimated 58 million people worldwide, with 1.5 million new infections annually. Chronic hepatitis C affects approximately 3.2 million adolescents and children, underscoring its pervasive impact [1]. Despite the efficacy of direct-acting antivirals (DAAs), which cure over 95% of HCV cases, access to diagnosis and treatment remains limited, and the lack of an effective vaccine adds to the complexity.

HCV is a single-stranded, positive-sense RNA virus within the Flaviviridae family [2]. Its RNA genome of approximately 9600 nucleotides encodes a single polyprotein that is processed by viral and cellular proteases into three structural proteins (core and envelope glycoproteins E1 and E2), as well as seven nonstructural proteins (p7, NS2, NS3, NS4A and B, and NS5A and B). The 5′ non-coding region (NCR) of the viral RNA genome and the first 12–30 nucleotides of the core region comprise the internal ribosome entry site (IRES) spanning nucleotides 39 to 371, which is essential for initiation of protein synthesis [3]. 

HCV displays remarkable genetic variability, with 8 distinct genotypes and more than 90 subtypes identified [4,5]. The high mutation rates of the virus, driven by RNA-dependent RNA polymerase, and the selective pressure of the host immune system result in a dynamic “viral population” within infected individuals. It is known, however, that substitution rates differ among the various HCV genes, and it has been observed that the 5′NCR (which harbors most of the IRES) is much more conserved than other parts of the genome [6,7]. This region evolves at the slowest rate, while the polyprotein precursor evolves on average about seven times faster [8]. This, together with the importance of the IRES for viral translation, makes it the most conserved region in the HCV genome. Both the secondary structure and its primary sequence play fundamental roles in translation initiation and, if altered, can affect translation activity [9,10,11].

Viral recombination is the process responsible for creating chimeric genomes by combining segments from different phylogenetic origins. This phenomenon greatly contributes to the genetic variation of RNA viruses and requires co-infection with two or more viral variants to occur. However, natural recombination events are rare in HCV, probably due to the fact that infected cells are less permissive for replication of a second infecting virus, making infection of a single cell by multiple HCV strains unlikely [12,13]. Nevertheless, several types of recombination have been reported over the years and have been categorized into three types: inter-genotype, inter-subtype, and intra-patient/intra-subtype recombination [14]. Previous research has identified eleven inter-genotype recombination types (2k/1b, 2i/6p, 2b/1b, 2/5, 2b/6w, 4/1, 3a/1b, 3a/1a, 2b/1a, 2a/1a, and 6a/3a) and seven inter-subtype recombination types (1b/1a, 1a/1c, 6a/6o, 6e/6o, 6e/6h, 6n/6o, and 4a/4d) based on either full-length or partial genome sequences [14,15,16,17,18,19,20,21,22,23,24,25,26,27,28,29,30,31,32,33,34]. Intra-patient recombination has also been reported [35,36], although it is more difficult to detect due to the genetic relatedness of the viral sequences. An important point to mention is that most inter-genotype and inter-subtype recombination events have been detected when analyzing consensus sequences, whereas intra-patient recombinant detection has relied upon molecular cloning and/or next-generation sequencing, and in all cases evidenced favored intra-subtype recombination [37]. Breakpoints for recombination have been identified in various regions of the HCV genome, including non-structural and structural domains [14,15,16,17,18,19,20,21,22,23,24,25,26,27,28,29,30,31,32,33,34]. While the full extent of recombination impact remains to be determined, it has been suggested that recombination accelerates viral adaptation by bringing together multiple advantageous mutations that may provide drug resistance or escape that may have profound implications for diagnosis, therapy, and epidemiology [38].

Next-generation sequencing methods offer a promising alternative to traditional sequencing approaches in the study of viral genetic diversity. These advanced techniques provide a comprehensive view of viral populations, enabling the detection of mixed infections and the identification of recombinant variants, even when present as minority or low-frequency variants [39,40,41,42]. 

In this study, we report two cases of mixed infection with genotypes 1a (Gt1a) and 3a (Gt3a), inferred by deep sequencing and verified by molecular cloning. In addition, this is the first report of 1a/3a and 3a/1a inter-genotypic recombinant forms within the IRES region, observed as low-frequency variants, detected in one of the dual-infected patients.

## 2. Materials and Methods

### 2.1. Clinical Samples and Ethical Considerations

Serum samples were collected from 43 patients chronically infected with HCV. All patients were adults (18 years old or older) and serologically negative to hepatitis B virus. Samples were collected between 2012 and 2016 in two different hospitals from Montevideo, Uruguay (Asociación Española—private setting, and Hospital de Clínicas—public university hospital). All studies were performed according to national and international ethical guidelines (good clinical practices and the Declaration of Helsinki) and were approved by the corresponding institutional boards. Written informed consent was obtained from all patients. Their personal information is confidential, and access was restricted to medical personnel only. Demographic data, such as sex and age, as well as virological parameters at the time of sample collection (viral load, antiviral therapy, and response), were also registered (Appendix A). 

### 2.2. Genotype Determination

In order to determine the genotype of each HCV isolate, serum samples were subjected to RNA extraction using the QIAamp Viral RNA Mini Kit (QIAGEN, Hilden, Germany), followed by cDNA synthesis using Superscript II reverse transcriptase (Invitrogen Life Technologies, Carlsbad, CA, USA), according to the manufacturers’ instructions. Next, a 386 bp conserved region from the NS5B polymerase gene was amplified by a hemi-nested PCR protocol using primers previously described [43]. Second-round PCR products were visualized on 2% agarose gels stained with SYBR Safe DNA gel stain (Invitrogen, Madison, USA), purified using the QIAquick PCR Purification Kit (QIAGEN, Hilden, Germany) and sent to Macrogen Inc. (Seoul, Republic of Korea) for bidirectional sequencing. Sequences were edited in the Seqman software version 7.0.0 implemented in DNAStar 5.01 (DNASTAR, Madison, WI, USA) and subsequently aligned to reference sequences of all 8 HCV genotypes using Clustal W algorithm [44] implemented in Mega11 software, version 11.0.13 [45]. A total of 40 reference sequences of complete genomes were downloaded from the Los Alamos Hepatitis C sequence database [46] (Appendix A). A Maximum Likelihood phylogenetic tree was constructed using Mega11 software [45], including 500 bootstrap pseudo-replicates and using the best evolutionary model that fit the sequence data through the Akaike Information Criterion. The genotype of each HCV strain was inferred according to the clades and tree topology. 

### 2.3. One-Step PCR for IRES Amplification

A one-step PCR reaction for IRES amplification was performed using primers that amplify almost the entire 5′-NCR region and the first nucleotides of the gene encoding the core protein, thus covering the entire IRES region and beyond (370 base pairs (bp), from nt 14 to 383). PCR amplification of the IRES region was performed using primers previously described [10]. The reaction mixture for the one-step PCR was prepared in a final volume of 30 μL containing 15 μL of buffer (2×, containing: magnesium sulfate (MgSO_4_) at a final concentration of 1.2 millimolar (mM), and deoxynucleotide triphosphates (dNTPs) at a final concentration of 200 micromolar (µM)), 1 μL each of primers (10 μM), 0.2 μL of Ribolock (40 U/μL, Thermo Fisher, Vilnius, Lithuania), 0.6 μL of SuperScript III/Taq Platinum High-Fidelity retro-transcriptase enzyme mix (Invitrogen, Carlsbad, CA, USA), 7.2 μL of DEPC water, and 5 μL of RNA. RT-PCR cycling consisted of incubation at 48 °C for 45 min (retro-transcription (RT)), followed by pre-denaturation at 95 °C for 5 min and 40 cycles of 95 °C for 30 s (s), 58 °C for 30 s, 68 °C for 30 s, and a final extension at 68 °C for 10 min. The generated amplicon has an expected size of 391 bp since primers incorporate restriction recognition sites. IRES products were visualized in 2% agarose gels stained with SYBR Safe DNA gel stain (Invitrogen, Madison, WI, USA), purified, and sent for bidirectional sequencing, as described for the NS5B region. Sequences were also edited in the Seqman software implemented in DNAStar 5.01 (DNASTAR, Madison, WI, USA) to obtain the consensus sequence for each IRES.

### 2.4. Analysis of Intra-Host IRES Population Diversity by NGS and Molecular Cloning

Within the 43 original samples, we performed NGS on 2 samples, 1 assigned to Gt1a (1021) and 1 to Gt3a (05), for which Sanger sequencing showed double peaks in key positions of the IRES region. Those positions correspond to polymorphic sites between genotypes 1a and 3a. In these two samples, we also performed the analysis of intra-host IRES population diversity by molecular cloning.

For next-generation sequencing, libraries were prepared using the TruSeq Nano DNA LT Library prep kit (Illumina, San Diego, CA, USA), following the manufacturer’s instructions. IRES PCR products obtained by one-step PCR were purified and quantified using the Qubit 2.0 Fluorometer dsDNA HS Assay (Life Technologies, Eugene, OR, USA) before use. A paired-end (2 × 250) sequencing run was performed with the MiSeq Reagent Kit v2 (500 cycle; Illumina, San Diego, CA, USA) on a MiSeq instrument (Illumina) from the sequencing service of Institut Pasteur Montevideo. Data pre-processing, alignment, and visualization were performed in the Galaxy web platform http://usegalaxy.org (accessed on 13 March 2019) [47]. Briefly, raw data were analyzed with the FastQC program [48], and the Trimmomatic tool was used to remove contamination sources [49]: a sliding window (4 nt) algorithm was used to remove adapters, primer sequences, and all reads with an average quality score below 25 (Q < 25); then, reads shorter that 36 nt were removed by the MINLEN algorithm. Quality control was performed with the FastQC program. As reference sequences for assembly, IRES consensus sequences previously obtained by the Sanger methodology were employed. The software BWA-MEM version 0.7.17.1 [50] was used to map good-quality reads to the consensus sequences. With the Qualimap 2 tool, we evaluated the coverage profile of each sample [51]. The algorithm MPileUp from SAMTools was used for variant calling [52]. All nucleotide positions with variants’ frequencies below 15% were analyzed. 

For cloning, we used the pCR™2.1-TOPO™-TA Vector (Invitrogen, Carlsbad, CA, USA) and chemically competent Top10 *E. coli* bacteria. Since our IRES PCR products were generated with a high-fidelity polymerase and lacked 3′ overhangs, products were adenylated and purified before the transformation protocol. Then, 1–4 µL of purified IRES was mixed with 1 µL of TOPO vector and water to a final volume of 6 µL and, after 5 min incubation at room temperature, 2 µL was mixed with a vial of Top10 bacteria and incubated for 10 min on ice before heat shock for 30 s at 42 °C. Next, 250 µL of SOC medium was added, and the tubes were incubated with shaking for 1 h. Then, 50 µL from each transformation were spread on prewarmed LB plates containing ampicillin and X-gal and were incubated overnight at 37 °C. We then picked around 60 white colonies both for colony PCR insert verification as well as for subculturing in 96-well plates with LB agar plus ampicillin to be sent for PlateSeq sequencing to Eurofins Genomics (Ebersberg, Germany). Sequences were then edited in the Seqman software implemented in DNAStar 5.01 (DNASTAR, Madison, WI, USA) and compared to 1a or 3a reference sequences (strain H77—Accession No. AF009606 and strain NZL1—Accession No. D17763, respectively) to analyze intra-host genetic variability. 

### 2.5. Recombination Analysis

To identify possible recombination events, those clones that were suspected to be recombinants due to nucleotide changes in polymorphic sites were subjected to further recombination analyses. 

The potential recombination events were detected by SimPlot++ software V1.3 [53]. This program is based on a sliding window method and constitutes a way of graphically displaying the coherence of the sequence relationships over the entire length of a set of aligned homologous sequences, using a window length of 100 bp, a step size of 10 bp, and a GTR-optimized distance model. In all cases tested, query sequences corresponded to potential recombinant clones (A33, B12, and C4) analyzed against IRES reference sequences of 1a and 3a genotypes (strains H77 and NZL1, see the previous section) as potential parental sequences. The results obtained in the recombination analysis by Simplot++ software were confirmed using a Bootscan analysis, as implemented in the Simplot Program Version 3.5.1 [54]. The window width and the step size were also set to 100 bp and 10 bp, respectively, whereas the distance model used was Kimura (two-parameter model).

The candidate recombination events were additionally verified by the Recombination Detection Program (RDP4) package version 4.101 [55] with the application of at least three of six statistical methods. The methods implemented included the original RDP method [56], GENCONV [57], Bootscan [58], SISCAN [59], LARD [60], and 3SEQ [61]. 

## 3. Results

### 3.1. Mixed Infections Were Detected in Chronic HCV-Infected Patients, Initially Inferred by Deep Sequencing and Then Verified by Molecular Cloning 

With the aim of analyzing IRES variability in chronic HCV-infected patients, we aligned and compared the full IRES regions obtained by PCR from each of the 43 samples to their respective reference sequences of each genotype. Since that region is the most conserved within the genome and, therefore, not suited for phylogenetic analyses and genotype determination [62], we used a partial region of the NS5B gene for genotype assignment. 

Of the 43 samples analyzed, 26 were assigned to genotype 1a (Gt1a), 12 to genotype 1b (Gt1b), and 5 to genotype 3a (Gt3a) according to NS5B phylogeny (see Appendix A). 

Once the samples were assigned to their genotypes, we were able to determine the IRES variability by comparing each sequence with reference sequences. To do this, we first processed the chromatograms obtained by Sanger sequencing, which led us to suspect the presence of mixed infections in two patients (samples: 1021—Gt1a and 05—Gt3a) by detecting small nucleotide peaks at key positions that distinguished genotypes 1a from 3a at the IRES level (see Appendix A). 

Then, in order to confirm the presence of possible recombination events, we conducted an analysis of intra-host IRES population diversity on these two IRES samples using both next-generation sequencing (NGS) and molecular cloning. 

Using NGS methods, we observed differences between majority and low-frequency variants (1–15% of reads) of the viral populations in the two samples analyzed. These differences were located at 19 specific nucleotide positions in the non-coding region of the IRES that were characterized and differed between Gt1a and Gt3a. Sample 1021, assigned to Gt1a, seemed to exhibit nucleotides characteristic of Gt1a as majority frequency variants, but other variants present at frequencies of 1.0–1.9% corresponded to Gt3a. On the contrary, in sample 05, the variants present at frequencies of 6.7–8.7% corresponded to Gt1a, while the majority seemed to be Gt3a. Thus, these results hint at the existence of mixed infections between genotypes 3a and 1a (see Figure 1), in which the predominant viral genotype is the one detected at the level of the consensus sequences of the different regions analyzed by Sanger sequencing. 

To confirm that the changes detected in nucleotide frequencies by NGS actually represent mixed infections, we performed molecular cloning of IRESs from these two patients. In sample 1021, of the 54 clones analyzed, 53 were assigned to Gt1a and 1 clone sequence was assigned to Gt3a (representing 1.85% of the population). On the other hand, in sample 05, of the 54 clones analyzed, 50 were Gt3a and 1 was Gt1a. These results suggest the existence of mixed infections in these two patients. Moreover, nucleotide mutations found in three clones from sample 05 (clones B12, C4, and A33, 5.5% of the population) indicate the presence of recombinant sequences (see Figure 2A).

Alignment of these 3 clones revealed that there is a stretch of conserved nucleotides (position 98 to 174) in Gt1a and Gt3a, flanked by upstream and downstream polymorphic sites, which suggest a change in genotype (see Figure 2A). The conserved region includes part of domains II and III of the IRES secondary structure (Figure 2B).

### 3.2. Recombinant Forms of HCV Were Found in a Patient with Mixed Infection

In order to confirm these three clones (B12, C4, and A33) as possible inter-genotypic HCV recombinants, we analyzed the alignment datasets by Simplot++ V1.3 [53], Simplot Version 3.5.1 [54], and RDP4.101 [55] software. The results of these studies are shown in Figure 3.

The similarity plots generated in Simplot++ were carried out using IRES reference sequences for genotypes 1a and 3a (strains H77 and NZL1, respectively) as potential parental sequences. The plots for each putative recombinant indicate a recombination breakpoint around nucleotide position 125 (see Figure 3, left panels). In the case of clone B12, it seems to have a 5′ region of Gt1a, followed by a 3a genotype rest of the IRES (Figure 3A). The opposite seems to be true for the other clones (C4 and A33; Figure 3B,C). Moreover, the Bootscan graphs generated by Simplot software (version 3.5.1) indicated similar results (Figure 3, right panels). 

To confirm the results found, a full exploratory recombination scan was performed with RDP4.101 software. The results of these analyses revealed that clone B12 was a recombinant with statistical support obtained using six methods (RDP, GENECONV, Bootscan, SiScan, LARD, and 3Seq). Clones A33 and C4 were supported by three out of six methods: SiScan, LARD, and 3Seq (see Table 1). The RDP4 scan also suggested that the potential parent (Gt1a) for clones C4 and A33 is another clone found in the same sample (clone B2, Gt1a; Table 1). In addition, the recombination breakpoint predicted was around nucleotide position 136, within the conserved region previously observed (Figure 2A). UPGMA phylogenetic trees obtained with the RDP4.101 software for each portion of the IRES (nucleotides 41 to 135 and 136 to 359) clearly show the phylogenetic incongruence and suggest that clones C4 and A33 might, indeed, represent the same recombination event (Appendix A).

## 
4. Discussion


The results of this work, using advanced sequencing techniques such as deep sequencing and molecular cloning, revealed cases of mixed infections involving genotypes 1a and 3a, two of the most frequently detected genotypes in Uruguay [63,64]. These complementary methodologies not only verified the presence of mixed infections, but also revealed the occurrence of inter-genotypic recombinations 1a/3a and 3a/1a within the IRES region. 

Mixed infections, in which individuals are simultaneously infected with more than one HCV genotype (or subtype), have been documented in 5–10% of all HCV-positive patients in the general population [65] and seem to reach 14–39% among people who inject drugs [66]. However, the true prevalence of this phenomenon remains uncertain and is likely influenced by methodological bias. Although many strategies have been developed for HCV genotyping over time, current HCV genotyping methods may have some limitations in detecting mixed infections [41,67,68,69,70]. Next-generation sequencing techniques provide an alternative to traditional sequencing methods for studying viral genetic diversity. These advanced approaches provide a comprehensive view of viral populations, facilitating the detection of mixed infections [71] and the identification of recombinant forms, even when present as minority variants. Despite the advantages that new technologies bring to the detection of mixed infections, their use should be rigorous and their results cautious. A recent study comparing three different deep sequencing methods was able to determine the presence of mixed infections in 1 out of 143 samples analyzed, even when many reads detected at 1–2% mapped to different genotypes. However, most of those reads did not pass the mixed-infection cut-off criteria [42]. Our NGS results indeed showed nucleotide changes present in very low frequencies (1–1.9%), which were later confirmed by cloning (sample 1021, see Figure 1), which highlights the importance of combining both methods for accurately detecting mixed infections characterized by a large disparity in the abundance of the major and minor genotypes. 

The importance of screening for mixed HCV infection has evolved with advances in antiviral therapy. Initially, treatment with pegylated interferon-alpha and ribavirin or early genotype-specific DAAs posed a challenge because mixed infections could lead to treatment failure, as elimination of the most abundant genotype would allow the minor genotype to become dominant [39,40]. However, with the introduction of pan-genotypic DAA regimens, the emphasis on genotyping and detection of mixed infections has decreased. Recent real-world data suggest high success rates with pan-genotypic DAAs, even in the presence of mixed infections [72,73,74]. As these studies include few or no treatment-experienced patients, it is still unclear whether mixed infections have an impact on treatment outcome in this patient cohort. However, the impact of mixed infections on treatment outcomes in experienced patients, particularly those with genotype 3 as a minor strain and compensated cirrhosis, could be significant. These patients may require tailored treatment to ensure a sustained virologic response (SVR), possibly by adding ribavirin or extending the treatment duration [75]. In addition, the European Association for the Study of the Liver suggests that genotype and subtype identification by population or deep sequencing could be beneficial for migrants from regions where less treatment-susceptible HCV subtypes are prevalent. This approach allows detection of HCV subtypes that are inherently resistant to NS5A inhibitors, thereby reducing the risk of treatment failure [76]. Similarly, these recommendations are relevant for cases involving recombinant sequences comprising NS3, NS5A, and NS5B regions of genotypes that remain difficult to treat. In the two particular cases reported in this work, the importance of the detection of mixed infections is unclear. On the one hand, data from patient 1021 are scarce, as the patient was lost to follow-up (Appendix A). On the other hand, patient 05 was naive at the time of sample collection (Appendix A) but presented with a presumably long-standing untreated infection (28 years). This could be the key in acquiring a mixed infection if the patient was co-infected with two different genotypes at different times in her life. This patient started DAA-based therapy in October 2023 (sofosbuvir plus daclatasvir for 24 weeks), but the treatment outcome is still pending. However, the major genotype in this case is the most difficult to treat (Gt 3a), so the duration therapy was tailored from the start. It is worth noting that there does not appear to be any particular patient characteristic that distinguishes those with mixed infections from those with non-mixed infections, although a more in-depth NGS analysis of other samples may provide a different scenario.

Recombination could serve as a critical driving force for the evolution and genetic diversity in RNA viruses [77,78,79]. Unlike in other RNA viruses, the occurrence of a natural recombination event within HCV is not common [80]. This is supported by the observation of superinfection exclusion, where an established virus infection prevents or interferes with subsequent infection by a second virus [12,13]. In addition, although productive homologous and non-homologous recombination has been reported in vitro for HCV, inter-genotypic recombination seems infrequent [81]. Nevertheless, since 2002, several naturally occurring inter-genotypic HCV recombinants have been reported [15,16,17,19,21,22,23,24,25,26,27,29,30,31,33]. In this work, we report two inter-genotypic recombination forms involving genotypes 1a and 3a. These recombinants exhibit two notable characteristics: firstly, the recombination breakpoints were identified within the conserved IRES region, and secondly, the recombinant forms were detected through cloning approaches and found to exist as low-frequency variants. 

As mentioned above, recombinant breakpoints within the HCV genome have been identified in both non-structural and structural regions, with a preference for occurrence between the core and NS2/NS3 regions, and specifically within the NS2/NS3 when the recombination event occurs between different genotypes [37]. To the best of our knowledge, our work and that of Qi et al. (2021) [25] are the only studies to show inter-genotypic recombinant breakpoints within the IRES region. This infrequent finding is somewhat expected considering the critical role of the IRES in viral translation and its high degree of conservation. However, it is worth noting that despite its conserved nature, the IRES may exhibit a higher degree of variability due to recombination than previously recognized. Our study allowed these inter-genotypic recombination events to be detected and verified thanks to the inclusion of molecular cloning methods, which have so far not been exploited to this end. Analyzing intra-host IRES variability by two complementary approaches (NGS and cloning) led us to not only the detection of mixed infections but, more importantly, the identification of IRES recombinant forms and their potential parental strains among the less frequent genotype. Analysis by molecular cloning has been the most frequent methodology used to study IRES variability and the translation efficiency of different IRES variants [82,83,84]; however, this is the first report to find mixed infections and recombinant forms of this short and highly-conserved HCV genome region. 

We also must consider the impact of selective pressure, which plays a critical role in determining the persistence or extinction of certain variants within the HCV IRES region. Recent research by Galli et al. (2022) [85] used a cell-culture-based assay to measure recombination events during replication of fully viable HCV infectious clones. They demonstrated that HCV can recombine at a high frequency even in the absence of external selective pressure, at least between highly similar genomes. Their findings suggest that the low incidence of natural recombinant forms is more likely the result of strong selective forces acting against most recombinants rather than a limitation in the inherent recombinant capacity of HCV itself. This argues in favor of the observation that most genetic cross-overs detected in HCV sequences isolated from patients occur in the NS2–NS3 junction region, which might indicate that recombination involving entire structural and non-structural genomic fragments is more likely to yield viable and fit viruses. However, one limitation of their study is the use of highly related genetic clones (all corresponding to genotype 2a), which might have led to an overall overestimation of HCV in vitro recombination ability. Even so, the high intrinsic recombination capacity of HCV seems to be supported by reports of frequent detection of intra-patient (intra-subtype) recombinant strains, even in the absence of superinfection scenarios where recombinant selection pressure would be minimal due to genetic similarity between strains [36]. This situation might argue in favor of frequent homologous recombination due to the genetic relatedness of the sequences circulating within a host.

Within the IRES region, the constraints imposed both by functional and structural requirements are likely to be more stringent, potentially resulting in less efficient IRES function and, consequently, a lower prevalence of observed inter-genotypic recombinants as major sequences within the viral population. It can be hypothesized that continuous selection pressure in vivo results in an overall lower frequency of IRES recombinants. This may explain why only one inter-genotypic recombinant form has so far been detected in a consensus sequence derived from a patient (6a/3a) [25]. Our report is the first to detect 1a/3a and 3a/1a IRES recombinant forms, albeit as low-frequency variants within the viral intra-host population. Since the putative cross-over point is located within a very conserved sequence stretch of the IRES, it is feasible that homologous recombination around this region is more common than it has been reported so far due to a lack of screening. This idea is somewhat supported by our finding of three recombinant clones in the same patient sample, accounting for 5.5% of the viral population. Whether these new recombinant forms are translationally efficient remains to be elucidated. 

Our study has a few limitations that warrant further research. First, the analysis of the two samples by NGS and subsequent cloning might have resulted in underestimating the frequency of mixed infections and recombination events in Uruguayan patients. Second, as we analyzed only the IRES region, we cannot exclude the possibility of multiple recombination events in the same clones, as well as other recombination sites along the viral genome in other clones. Lastly, since our study was exploratory and descriptive, we did not perform mechanistic experiments to assess the functionality of the identified IRES recombinant forms. 

In conclusion, using NGS and molecular cloning, we confirmed the presence of mixed infections in two chronic HCV-infected patients from Uruguay. In addition, in one of them, we detected two inter-genotypic recombinant forms within the conserved IRES region (1a/3a and 3a/1a) in 5.5% of the viral population. This is the first report of 1a/3a and 3a/1a HCV recombinant forms within the IRES. Our findings suggest that the conserved IRES region may have a higher degree of variability due to recombination than previously reported.

## Figures and Tables

**Figure 1 viruses-16-00560-f001:**
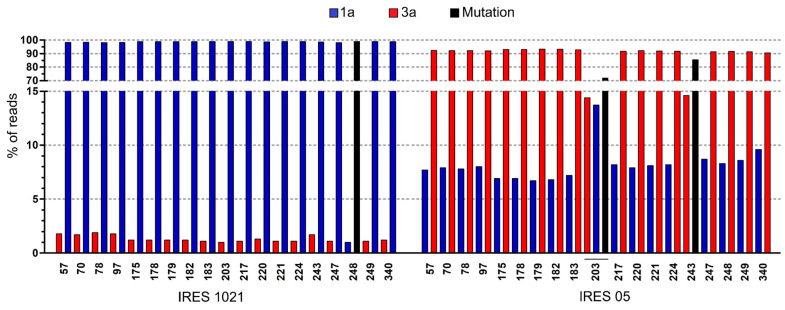
Majority and low-frequency variants identified in the viral populations of IRES from samples with suspected mixed infections. IRES 1021 and 05, from Gt1a and 3a samples, respectively, are shown. Nucleotide positions that are different among genotypes 1a and 3a in the non-coding region of IRES are shown. Both low-frequency (1–15% of reads) and majority changes are represented: in blue, those specific to genotype 1a, in red, those specific to genotype 3a, and in black, those accounting for mutations (nucleotides different from those of the reference of one or the other genotype).

**Figure 2 viruses-16-00560-f002:**
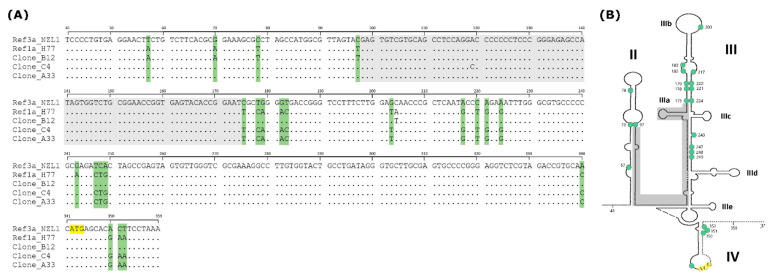
Alignment of IRES clones corresponding to sample 05 (genotype 3a) suspected to be recombinant sequences and a putative cross-over region. (**A**) Full IRES region is shown, including 5′NCR from position 41 onwards, and the core region up to position 18 (nucleotide 359 of the genome). Numbering is according to strain H77 (reference sequence for genotype 1a). All positions indicated with dots correspond to conserved nucleotides. Those highlighted in green indicate polymorphic sites between genotypes 1a and 3a within the 5′NCR or the core regions. Positions 98 to 174 (highlighted in grey) indicate a conserved sequence stretch, where the putative cross-over occurred (clone B12 changes from genotype 1a to 3a, whereas C4 and A33 change from 3a to 1a). The start codon is indicated in yellow. (**B**) HCV IRES secondary structure indicating the same features as in (**A**). IRES domains (II, III, IV) and subdomains are indicated with roman numbers (IIIa to IIIe).

**Figure 3 viruses-16-00560-f003:**
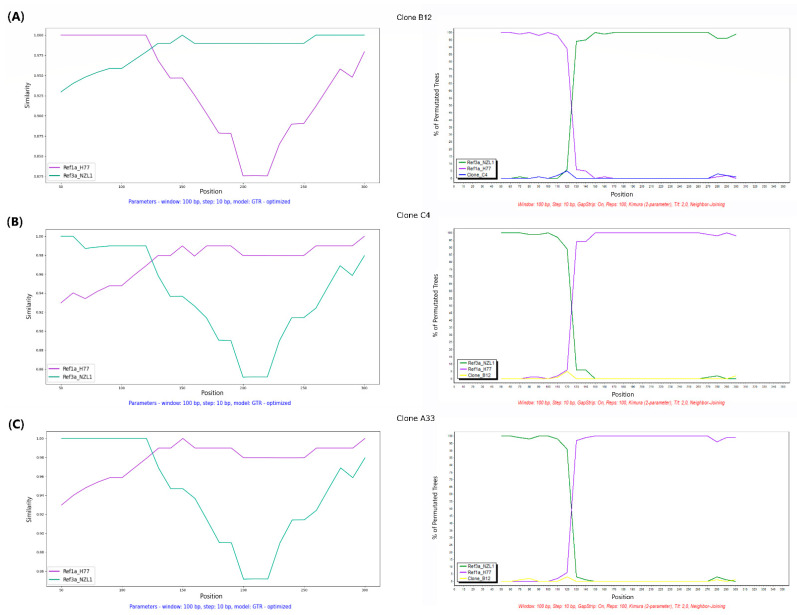
Similarity plots and Bootscan analyses of potential HCV recombinants determined on the basis of the IRES complete sequence using Simplot version 3.5.1 [54] and Simplot ++ V1.3 [53] software. Simplot++ analyses (left panels) display the percentage of sequence similarity over the IRES sequences of the HCV genome (nucleotide positions 41 to 359 according to strain 1a H77 numbering). Bootscan analysis (right panels) shows the percentage of permuted trees (y-axis). This approach permits to observe levels of phylogenetic relatedness between a query sequence and a reference sequence in different genomic regions. Query sequences (shown on the upper part of the figure) correspond to (**A**) clone B12 (putative 1a/3a recombinant), (**B**) clone C4 (putative 3a/1a recombinant), and (**C**) clone A33 (putative 3a/1a recombinant). Reference and/or putative parental sequences are indicated in purple for genotype 1a (strain H77, NCBI Accession No. AF009606) and green for genotype 3a (strain NZL1, NCBI Accession No. D17763). All analyses were performed using a 100 bp sliding window and a 10 bp step size. For Simplot++ analyses, a GTR optimized model of nucleotide substitution was employed, whereas for Bootscan, a Kimura two-parameter model was used.

**Table 1 viruses-16-00560-t001:** Recombination events detected by RDP4 in HCV IRES clones from sample 05.

			Detection Methods and *p*-Values					
Recombinant Sequence(s)	Minor Parent	Major Parent	RDP	GENECONV	Bootscan	SiSscan	LARD	3Seq
Clone_B12	Ref1a_H77	Ref3a_NZL1	4.249 × 10^−2^	1.760 × 10^−2^	5.260 × 10^−3^	5.037 × 10^−6^	9.215 × 10^−3^	6.835 × 10^−4^
Clone_C4	Clone_B2	Ref3a_NZL1	NS	NS	NS	3.472 × 10^−10^	5.20 × 10^−3^	8.322 × 10^−3^
Clone_A33	Clone_B2	Ref3a_NZL1	NS	NS	NS	7.802 × 10^−14^	1.212 × 10^−2^	8.322 × 10^−3^

## Data Availability

The sequences in this study are available in the GenBank database (NS5B partial sequences—accession numbers PP239648 to PP239690, IRES sequences—accession numbers PP350539 to PP350580, Sample 05 IRES clones—accession numbers PP350581 to PP350634, Sample 1021 IRES clones—accession numbers PP350635 to PP350688). NGS data are available under Bioproject accession number PRJNA1078827.

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
