# Peer review of "Mixed Infections Unravel Novel HCV Inter-Genotypic Recombinant Forms within the Conserved IRES Region"

_viruses, 2024, doi:10.3390/v16040560_

Round 1

Reviewer 1 Report

Comments and Suggestions for Authors

1.  These authors have analyzed the genotypes of 43 HCV samples from Uruguay.  They used a region in the NS5B gene for genotype assignment. Of the 43 samples, 26 were assigned to genotype Ia, 12 to genotype 1b, and 5 to genotype 3a.  They were able to identify mixed infections in 2 patients.  In both cases there was a predominant genotype and a minority genotype.
3.  They then tried to identify possible recombination events using next generation sequencing and molecular cloning.  They were able to identify inter-genotypic HCV recombinants in one patient.
4.  This information has the potential to help understand HCV evolution and adaptation and potentially drug resistance.  They suggest that the IRES region may have more variability due to recombination than previously reported.

5.  These authors have written an excellent paper.  They provide a good introduction, detailed methods, comprehensive results, and a long discussion of the results and their implications.  Supplementary table 1 provides significant detail about the viral isolates in this study.  In addition, supplementary table 2 provides an overview of the worldwide distribution of HCV genotypes and the country in which the original isolation occurred. This paper should be of special interest to HCV virologists.

.

Author Response

We would like to thank the reviewer for his/her comments and appreciation of our work.

Reviewer 2 Report

Comments and Suggestions for Authors

Echeverría et al. present an interesting article detailing the discovery of a pair of recombinant viruses between genotypes 1 and 3 of the hepatitis C virus. The article is well-written, addressing most of the requirements regarding methodology and results description. The study's strength lies in identifying the breakpoint for recombination of both sequences, located in the IRES region. However, a notable weakness is that only a small segment of the genome is sequenced, limiting the ability to identify potential additional recombination points. Nevertheless, the study discusses this limitation, among others, at the end of manuscript.

It is important to clarify whether the samples used for sequencing were obtained from patients before, during, or after treatment.

To enhance the completeness of the experimental record, particularly as a reference for similar studies, I suggest including the electropherogram (as mentioned on line 226) as panel A in Figure 1.

Reviewer 3 Report

Comments and Suggestions for Authors

It is an interesting manuscript about “ Mixed Infections Unravel Novel HCV Inter-genotypic Recombinant Forms Within the Conserved IRES Region”.

My concern is determined in the following points.

The presence of mixed infections in two chronic HCV-infected patients from Uruguay. In addition, in one of them, authors detected 2 inter-genotypic recombinant forms within the conserved IRES region (1a/3a and 3a/1a) in 5.5% of the viral population. This is the first report of 1a/3a and 3a/1a HCV recombinant forms within the IRES. Our findings suggest that the conserved IRES region may have a higher degree of variability due to recombination than previously reported.

More detailed case presentation about two cases including clinical course, treatment, viral clearance and hepatic histological findings should be presented. Especially what differences were there compared with cases with non-mixed infection those.

Above mentioned should be referred to.
